# High-Molecular-Weight Hyaluronic Acid Can Be Used as a Food Additive to Improve the Symptoms of Persistent Inflammation, Immunosuppression and Catabolism Syndrome (PICS)

**DOI:** 10.3390/biology13050319

**Published:** 2024-05-03

**Authors:** Yuanyuan Jiang, Ye Jiang, Lu Li, Xiangyu Liu, Xiaoming Hou, Wenfei Wang

**Affiliations:** College of Life Science, Northeast Agricultural University, Harbin 150030, China; jyy580@126.com (Y.J.); yeri1025@163.com (Y.J.); lilu021302@163.com (L.L.); lily_liu1979@126.com (X.L.)

**Keywords:** PICS, hyaluronic acid, gut flora, food additives

## Abstract

**Simple Summary:**

Persistent inflammation, immunosuppression, and catabolic syndrome (PICS) are the main causes of poor outcomes in patients transferred out of ICU. Hyaluronic acid has a potential function to alleviate PICS, and, as a widely accepted food additive, HA can be consumed through food products such as dairy products, greatly increasing patient medical compliance. This study evaluated the remission effect of PICS model mice after oral administration of HA with different molecular weights. The results showed a nonlinear relationship between HA molecular weight and its effect of intervention with symptoms in PICS mice and the high molecular weight HA (HMW-HA) was more effective in relieving symptoms in PICS model mice. Therefore, we believe that HMW-HA can be used to prepare functional foods for the purpose of intervening with PICS disease.

**Abstract:**

Hyaluronic acid (HA) is a new functional food additive which has the potential to ameliorate persistent inflammation, immunosuppression and catabolism syndrome (PICS), but the biological effects of HA with various molecular weights differ dramatically. To systematically investigate the efficacy of HA in altering PICS symptoms, medium-molecular-weight (MMW) HA was specifically selected to test its intervention effect on a PICS mouse model induced by CLP through oral administration, with high-molecular-weight (HMW) and low-molecular-weight (LMW) HA also participating in the experimental validation process. The results of pathological observations and gut flora showed that MMW HA rapidly alleviated lung lesions and intestinal structural changes in PICS mice in the short term. However, although long-term MMW HA administration significantly reduced the proportions of harmful bacteria in gut flora, inflammatory responses in the intestines and lungs of PICS mice were significantly higher in the MMW HA group than in the HMW HA and LMW HA groups. The use of HMW HA not only rapidly reduced the mortality rate of PICS mice but also improved their grip strength and the recovery of spleen and thymus indices. Furthermore, it consistently promoted the recovery of lung and intestinal tissues in PICS mice, and it also assisted in the sustained restoration of their gut microbiota. These effects were superior to those of LMW HA and MMW HA. The experimental results indicate that HMW weight HA has the greatest potential to be an adjunct in alleviating PICS as a food additive, while the safety of other HAs requires further attention.

## 1. Introduction

Persistent inflammation, immunosuppression, and catabolism syndrome (PICS) is a multifaceted disease caused by immune disorders, and no effective treatment is available because of its complex pathogenesis [1,2,3]. The main clinical intervention is to regulate diet and nutrient intake [4,5]. Sepsis is a life-threatening medical condition caused by infections, burns, and other factors with a mortality rate of 25–30% [6,7]. Although the mortality rate of sepsis has dramatically decreased with advances in medical treatments [8], sepsis survivors have serious health problems, and the prognosis is poorer for those who experience PICS [9,10,11].

Hyaluronic acid (HA), which is also known as hyaluronate and hyaluronan, is a non-sulfated glycosaminoglycan and endogenous mucopolysaccharide. It is composed of repeating disaccharide units of β-1,4-D-glucuronic acid and β-1,3-N-acetylglucosamine. As an important component of the extracellular matrix, several preclinical and clinical experiments have shown good efficacy regarding the anti-inflammatory, wound healing, and antiangiogenic properties as well as the safety profile of hyaluronic acid [12]. In addition to clinical application, HA has been used in the food processing industry as an egg white substitute as early as 1942 [13]. To date, lots of countries including China, the European Union and the USA have accepted the usage of HA as a food ingredient or dietary supplement. However, limited research studies are available that have explored the functional benefits of HA in dairy and food applications. HA has been proved that this molecule has strong immunomodulatory activities and regulates the intestinal system [14,15]; thus, there is a huge potential for the oral delivery of HA through foods to gain health benefits. Moreover, as a substrate for gut microbiota, HA may have beneficial impacts on the host by modulating the composition of the gut microbiota [16]. High-molecular-weight (HMW) HA prevents its free transit through the intestinal wall, which suggests that the HMW HA may had a greater impact on the gut microbiota composition.

Interestingly, the functional attributes of products containing HA would still depend upon the MW the HA used [17]. Our previous study demonstrated significantly different effects of high-molecular-weight (HMW) HA and low-molecular-weight (LMW) HA on PICS. HMW HA was better than LMW HA in relieving inflammation, whereas LMW HA was superior to HMW HA in improving the gut microbiota composition after long-term use [18]. However, the lack of experimentation with medium-molecular-weight HA directly restricts the practical comprehensive application of HA. Thus, it aroused our interest to explore whether the medium-molecular-weight (MMW) HA has therapeutic advantages over the above HAs.

In the present study, we established a PICS mouse model in accordance with the method described by Amanda et al. [19]. Typical HMW HA (1600 kDA), MMW HA (100 kDA), and LMW HA (3 kDA) were administered to this mouse model. By observing changes in the physiological indicators, pathology, and gut flora, we compared the effects of HMW, MMW, and LMW HA on PICS mice and investigated the feasibility of HA as a functional food additive in the dietary regulation of PICS patients.

## 2. Results

### 2.1. Survival, Grip Strength, and Organ Coefficient

To evaluate the therapeutic efficacy of HA of different molecular weights in the mouse models of PICS, we examined changes in the survival, grip strength, and organ coefficients in each experimental group. As Figure 1 shows, the risk of death in PICS mice (Figure 1A) gradually decreased with the recovery time but was still present in the early stage. After HA treatment, the mortality of treated mice was reduced by various degrees in all groups. The organ coefficient results on day 10 showed that with the increase in HA molecular weight, the spleen index was closer to the WT group (Figure 1C). However, the grip strength test on day 30 (Figure 1B) showed that the molecular weight of HA was positively correlated to the recovery of grip strength in mice. Further experiments are needed to determine the reason for this result and to assess the health status of mice.

### 2.2. Pathological Changes in Lung Tissues

We examined the lung lesions in the mice by using pathological sections. The results showed that short- and long-term pathological changes in the lungs of PICS mice were observed (Figure 2). The CK group had progressively deeper lung lesions with a longer disease duration, together with a disorganized lung architecture and remarkably thickened alveolar walls, and some alveoli had disappeared. Moreover, a large number of inflammatory cells infiltrating into alveolar and interstitial cavities were visible, which was accompanied by notable pathological features such as alveolar hemorrhaging and alveolar wall rupture. Mice in the 3 kDA-HA group showed a small amount of alveolar hemorrhaging and slight thickening of the alveolar wall in the short term. However, they still had more inflammatory cell infiltration in the long term. Conversely, mice in the 100 kDA-HA group did not have severe lesions in the short term. However, lung lesions worsened after prolonged administration of 100 kDA-HA along with a massive infiltration of inflammatory cells and tissue hemorrhaging. Mice in the 1600 kDA-HA group had more normal findings with no large or obvious lesions observed.

### 2.3. Morphological and Pathological Changes of Gut Tissues

Pathological changes in the gut were also the focus of examination in this study. After 10 days of HA administration, three mice were randomly selected from each group, and their intestines were harvested and observed for morphology (Figure 3). The intestinal system of mice in the CK group had pronounced atrophy and was more prone to rupture. Additionally, multiple swellings were observed in the intestinal wall, and the intestines tended to be black, which was probably because of long-term under-perfusion of the intestinal system or intestinal obstruction. Although HA groups also had swellings in the intestinal wall, the gut was essentially smooth and intact, and its color was similar to that in the WT group.

On days 10 and 30, the proximal and distal ends of the colon were harvested for H&E staining (Figure 3). In the short term, the CK group showed an atrophy of the lamina propria with a significant decrease in thickness, a decrease in the number of goblet cells and absorptive cells in crypts, and pronounced destruction of the upper structure of crypts. Conversely, the 3 kDA-HA group lacked the semilunar fold. After 30 days of HA administration, the intestinal structure had remarkably recovered in the 3 kDA-HA group, but it continued to deteriorate in the CK group. The 1600 kDA-HA group showed good recovery, which was superior to that in the other groups after day 10, and the intestinal structure subsequently became similar to that of the WT group. Changes in the intestinal structure were most notable in the 100 kDA-HA group. Recovery of the intestinal structure was far superior in the 100 kDA-HA group compared with the other groups on day 10. However, the intestinal structure in the 100 kDA-HA group showed significant inflammatory cell infiltration on day 30 along with disruption of the tissue structure.

### 2.4. Number and Diversity of Intestinal Flora

To clarify the correlation of the effect of HA molecular intervention in PICS and changes in gut microbiota, sequencing results were analyzed by clustering and taxonomy. The number of flora in the 1600 kDA-HA group gradually returned to normal, but the number of operational taxonomic units in the 3 kDA-HA and 100 kDA-HA groups was significantly lower than that in normal mice and lower than those in the other two groups of PICS mice (Figure 4A). Additionally, alpha-diversities in the 3 kDA-HA and 100 kDA-HA groups were closer to that in the WT group compared with the 1600 kDA-HA and CK groups (Figure 4B).

Beta-diversity analysis was performed on the flora composition and phylogenetic information of multiple samples (Figure 4C). During the initial period, the intestinal flora of all four PICS groups differed significantly from that of normal mice. However, the flora composition of the 3 kDA-HA, 1600 kDA-HA, and CK groups tended to recover toward the flora composition of the WT group. Meanwhile, the flora composition differed significantly between the 100 kDA-HA and WT groups.

### 2.5. Structure and Abundance of Gut Microbiota

Changes in the proportion and abundance of gut flora were most pronounced in the 100 kDA-HA group (Figure 5). The proportions of harmful bacteria were significantly reduced, including *Escherichia Shigella* in class Gammaproteobacteria, *Streptococcaceae* in class Bacilli, *Blautia* in class Clostridia, and *Erysipelotrichaceae* in class Erysipelotrichia. However, the proportions of beneficial bacteria, especially those involved in regulating the immune response (e.g., *Ruminococcaceae* and *Clostridiaceae_1* in class Clostridia) were significantly increased. The proportions of gut microbiota in the other groups were not significantly different from those in a previous study [18].

## 3. Discussion

PICS is a multifaceted disease involving immune system dysfunction and has the symptoms of metabolic and gastrointestinal disorders [20], which makes it difficult for patients to absorb and utilize even if they were given an adequate and nutritious diet, resulting in further deterioration of the patient’s condition [4,5]. It is difficult to treat this symptom with drugs. Enteral nutrition has been proved as a promising intervention to modulate the immune response and improve PICS [21]. Thus, the search for suitable complementary foods or food additives with the potential for ameliorating PICS holds greater feasibility and safety than the search for suitable therapeutic drugs.

HA regulates the immune system and improves the intestinal system. Therefore, HA may be a food additive that can regulate the health status of PICS patients. However, as the effects of HA with various molecular weights differed significantly, which molecular weight HA can be used as a food additive for improving PICS still needs further research. Our previous study found that HMW HA repaired the intestinal barrier and increased the stability of the intestinal flora composition in PICS mice, and oligomeric HA suppressed the progression of PICS mainly by regulating the intestinal flora. However, the effect of MMW HA on PICS remained unclear, which is a limitation that hinders the comprehensive analysis and practical application of the effects of HA on altering PICS symptoms. Therefore, we wonder whether MMW HA, whose molecular weight is somewhere in between HMA and oligomeric HA, can have the advantages of both. Thus, in this study, we explored the effect of 100 kDA-HA (typical of MMW HA) on the pathogenesis of PICS in mice, compared the effect of 100 kDA-HA with those of the other two HAs (3 kDA-HA and 1600 kDA-HA), and analyzed the effect of molecular weight on the efficacy of HA for PICS treatment.

The gut is an important site of action for oral HA and a vulnerable system in PICS patients [20]. In our study, H&E staining showed that the effects of 1600 and 3 kDA-HA on the gut were consistent with our previous findings. Specifically, 1600 kDA-HA significantly alleviated intestinal lesions within a short period, whereas 3 kDA-HA required a longer treatment course to improve the mouse colon structure. However, the effect of 100 kDA-HA was significantly different from those of HAs with other molecular weights. After short-term administration, 100 kDA-HA was superior to the other two Has in restoring colonic structures in PICS mice. However, after long-term administration, colonic structures worsened with more pronounced inflammatory cell infiltration. Therefore, long-term MMW HA administration may have promoted the inflammatory response. Because intestinal bacteria are the main inhabitants of the gut and the main contributors to intestinal immunity, their changes can, to some extent, reflect alterations in the immune environment in the gut [22,23,24]. Therefore, in our study, we examined the proportions and abundance of intestinal flora after long-term HA administration by 16S rRNA sequencing and analyzed their changes to verify our assumptions.

16S rRNA sequencing showed that although the proportions of harmful bacteria that are easily recognized and cleared by the immune system (e.g., *Escherichia Shigella* in class Gammaproteobacteria, *Streptococcaceae* in class Bacilli, *Blautia* in class Clostridia, and *Erysipelotrichaceae* in class Erysipelotrichia) [25,26,27] were significantly decreased in the 100 kDA-HA group, the proportions of bacteria involved in activating the immune response in the gut (e.g., *Ruminococcaceae* and *Clostridiaceae_1* in class Clostridia [28,29]) were significantly higher than those in the other two groups. Additionally, both the 100 kDA-HA and 3 kDA-HA groups had significantly lower flora abundances than the 1600 kDA-HA and CK groups. A possible explanation is that 100 kDA-HA and 3 kDA-HA over-activate the intestinal immune response, thereby reducing the abundance of intestinal flora in mice.

Consistent with the intestinal observations, lung pathology in mice also confirmed that the short-term recovery of lung structures in the 100 kDA-HA group was inferior to that in the 1600 kDA-HA group, but it was superior to those in the 3 kDA-HA and CK groups. However, pulmonary inflammation persisted in the 3 kDA-HA and 100 kDA-HA groups after long-term HA administration, especially in the 100 kDA-HA group. Nevertheless, the long-term administration of 100 kDA-HA did not lower the survival rate of mice.

MMW HA can be used as a raw material to synthesize large molecules or be broken down into small molecules [30,31]. Therefore, MMW HA may possess more intricate mechanisms of action compared to HMW HA and LMW HA. While we have not conducted experimental validation in this regard, we offer speculative hypotheses. When the effects of 100 kDA-HA and 1600 kDA-HA were compared, 1600 kDA-HA acts directly on intestinal tissues by repairing the intestinal barrier and regulating intestinal inflammation [32,33], which 100 kDA-HA did not. As a large molecule, 1600 kDA-HA is broken down in the body rather than being absorbed directly by patients [34]. Therefore, 1600 kDA-HA administration did not increase inflammation levels in PICS mice. Conversely, 100 kDA-HA is taken up by the body; then, it directly exerts its biological activities. In addition, a portion of 100 kDA-HA is decomposed in vivo into small molecules units for HA synthesis. Therefore, we speculate that after a prolonged 100 kDA-HA administration, the body breaks down large amounts of ingested exogenous MMW HA into smaller HA fragments, resulting in the accumulation of small molecular HA, which in turn activates the systemic inflammatory response and causes severe inflammatory cell infiltration in multiple tissues.

However, it is different from the degradation of 100 kDA-HA to produce large amounts of small pro-inflammatory molecules [35,36]; as an oligomeric HA, 3 kDA-HA has a small molecular weight and can be rapidly degraded or ingested by cells and does not form a continuous stimulation of the body’s immune system. Thus, it did not produce a pro-inflammatory effect similar to 100 kDA-HA in this study [34]. In some circumstances, 3 kDA-HA is used in the synthesis of LMW HA. HA of 3 kDA may also suppress the immune system by keeping patients in a state of high depletion, such as abnormal breakdown of muscle tissue, for reasons to be explored further.

It is noteworthy that hydrophilicity is also the most important property of HA molecules, which is the reason HA is widely used in food additives. The water-binding ability of HA can modify the viscosity and textural attributes of food. Thus, besides biomedical applications, the ability of HMW HA on binding with large amounts of water molecules also make it as a good food additive.

It should be pointed out that although the results of this study have clarified that high-molecular-weight HA can be used as a food additive in the rehabilitation treatment of patients with PICS, there are additional areas that need to be investigated. The gut microbiota contain enzymes capable of cleaving HA into shorter fragments; therefore, whether the effect of the treatment of HMW HA will change with time under the premise of long-term application still needs further study. Moreover, since the analysis level of gut microbiota in this study only reached the genus stages, further clarifying the effect of HA on the species stages of gut microbiota is still needed in the future to clarify the mechanism of HA regulating body immunity and substance metabolism.

In conclusion, while the short-term administration of 100 kDA-HA may alleviate PICS symptoms in mice, long-term 100 kDA-HA administration was associated with unfavorable outcomes. Additionally, 3 kDA-HA may also not be a suitable alternative to remission PICS. Conversely, 1600 kDA-HA improved the intestinal barrier while promoting recovery of the immune system. Therefore, the molecular weight of HA may determine its intervention effect on PICS, and HMW HA may be a potential food additive to alleviate the health status of PICS patients.

## 4. Materials and Methods

### 4.1. Modeling and Grouping

Healthy female Kunming mice (6 weeks old) were purchased from Liaoning Changsheng Biotechnology Co. Ltd. (Benxi, China). Mice were maintained in a controlled environment (23 ± 2 °C room temperature and 12 h/12 h light/dark cycle) with ad libitum access to water and normal chow. After acclimation for 1 week, mouse models of PICS were established in accordance with a previous study [37]. Fifty healthy mice were selected for cecal ligation and puncture (CLP), and 10 mice underwent a sham surgery (hereafter referred to as the WT group). The mortality rate of mice that underwent CLP was approximately 20% at 3 days after surgery, during which water and food were provided ad libitum. The surviving mice were randomly divided into four groups: model group (CK; *n* = 10), 3 kDA-HA group (3 K, *n* = 10), 100 kDA-HA group (100 K, *n* = 10), and 1600 kDA-HA group (1600 K; *n* = 10).

On postoperative day 3, CLP-operated mice surviving past the acute phase were used as PICS models for subsequent experiments. This time point was used as the start of the experiment (day 0), and mice in the 3 kDA-HA, 100 kDA-HA, and 1600 kDA-HA groups were orally administered the corresponding molecular weight of HA (30 mg/kg, Shandong Zhongshan Biotechnology Co., Ltd., Heze, China). Mice in the WT and CK groups were treated orally with an equal volume of saline. Oral administration started on day 3 after surgery (i.e., day 0 of the experiment) and continued until the end of the experiment.

We used day 10 of the experiment as a short-term observation node and day 30 as a long-term observation node.

### 4.2. Health Status Monitoring

The health status of mice was monitored throughout the treatment period. The survival status was recorded daily, and survival curves were plotted from day 0 (i.e., postoperative day 3) to the end of the experiment. Organ coefficient and grip strength were measured on days 10 and 30.

### 4.3. Histopathological Examination

On days 10 and 30, three mice from each group were randomly selected and sacrificed after anesthesia, and their lungs were harvested and fixed. Intestinal tissues were also collected. After morphological changes of the gut were observed, 1 cm sections were collected from proximal and distal ends of the colon and fixed separately. The fixed tissues were embedded in paraffin, sectioned, and stained with hematoxylin and eosin to observe histopathological changes.

### 4.4. Collection and Analysis of Gut Microbiota

Fresh excreta were collected on days 0 and 30 and subjected to 16S rRNA gene sequencing by Sangon Biotech (Shanghai) Co., Ltd. (China). The clustering numbers, diversity, and structural abundance of each taxon were analyzed, and the possible functions of these taxons in the gut were predicted.

### 4.5. Statistical Analysis

Statistical analyses were performed using GraphPad Prism 9.3.1 software. Data are presented as the mean ± standard error of the mean. The *t*-test was used for comparisons between groups. *p* < 0.05 was considered statistically significant.

## Figures and Tables

**Figure 1 biology-13-00319-f001:**
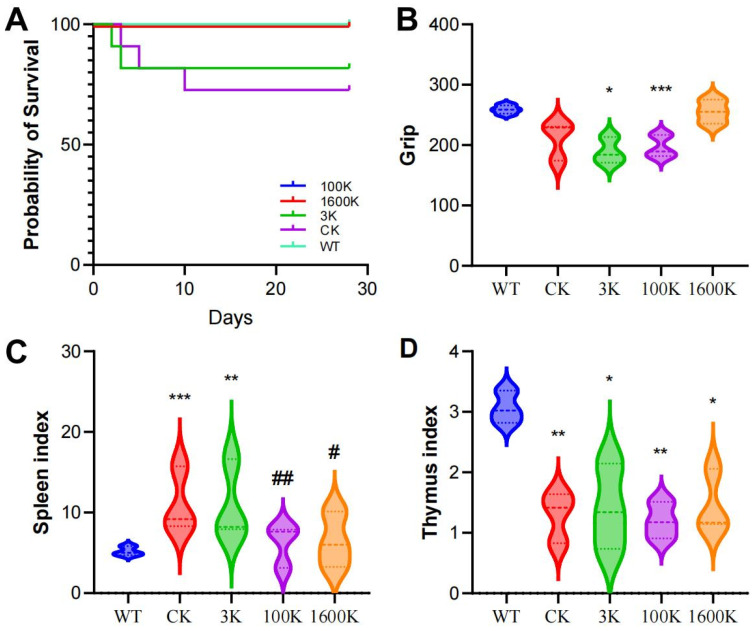
Effects of 3 kDA-HA (PICS mice model treated with 3 kDA-HA), 100 kDA-HA (PICS mice model treated with 100 kDA-HA), and 1600 kDA-HA (PICS mice model treated with 1600 kDA-HA) administration intervention on (**A**) survival rates; (**B**) grip strength; (**C**) spleen index; and (**D**) thymus index of PICS mice. Data are presented as mean ± standard deviation. * *p* < 0.05, ** *p* < 0.01 and *** *p* < 0.001 compared to WT group (mice underwent a sham surgery and treated with saline). ^#^ *p* < 0.05, ^##^ *p* < 0.01 compared to CK group (PICS mice model treated with saline).

**Figure 2 biology-13-00319-f002:**
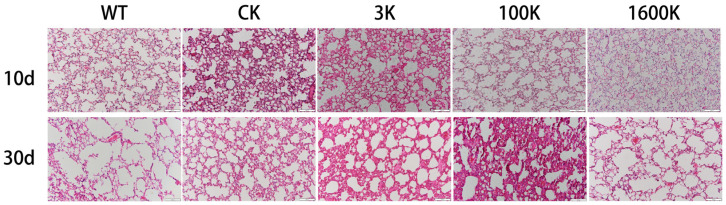
Results of H&E staining of PICS mice lungs on days 10 and 30 in each group after intervention with 3 kDA-HA, 100 kDA-HA, and 1600 kDA-HA administration. All representative histological images were taken at a magnification of ×200.

**Figure 3 biology-13-00319-f003:**
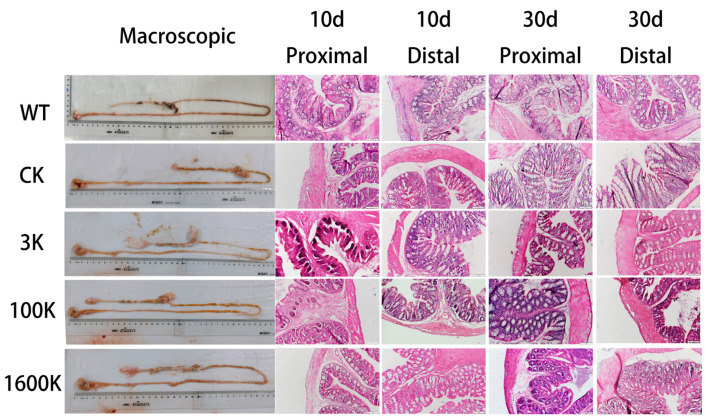
Results of macroscopic intestinal morphology at day 10 and H&E staining of intestinal tissues from PICS mice in each group at day 10 and 30 after intervention with 3 kDA-HA, 100 kDA-HA, and 1600 kDA-HA administration. All representative histological images were taken at a magnification of ×200.

**Figure 4 biology-13-00319-f004:**
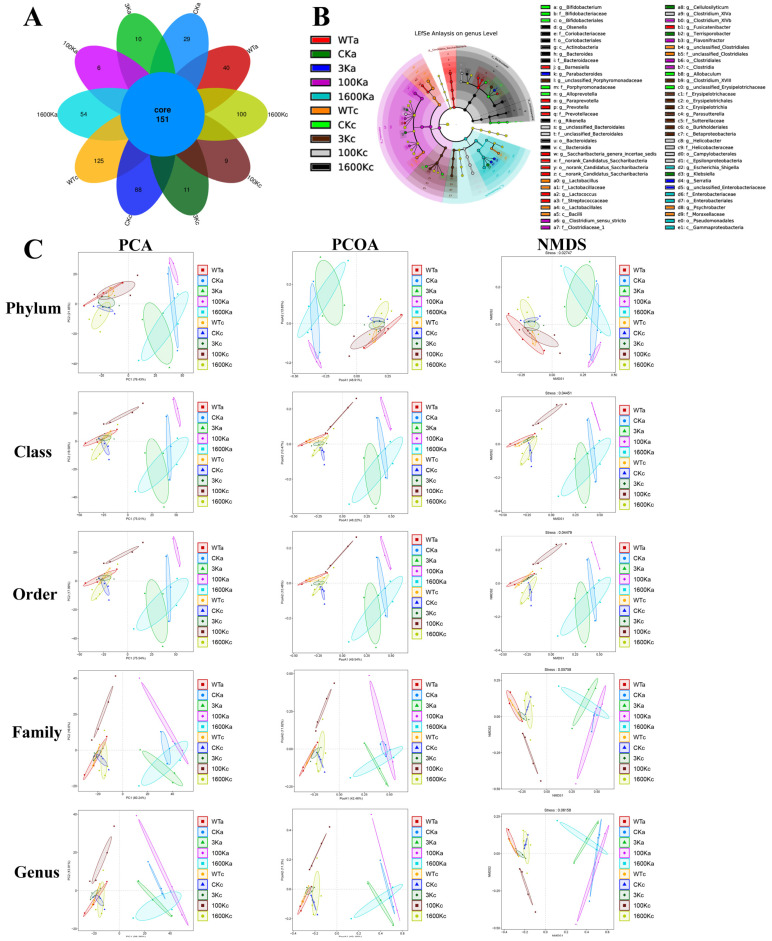
Effects of 3 kDA-HA, 100 kDA-HA and 1600 kDA-HA on the abundance and diversity of mice intestinal microbiota at day 30. (**A**) Venn diagram of microbial abundance; (**B**) microbial alpha diversity; (**C**) microbial beta diversity.

**Figure 5 biology-13-00319-f005:**
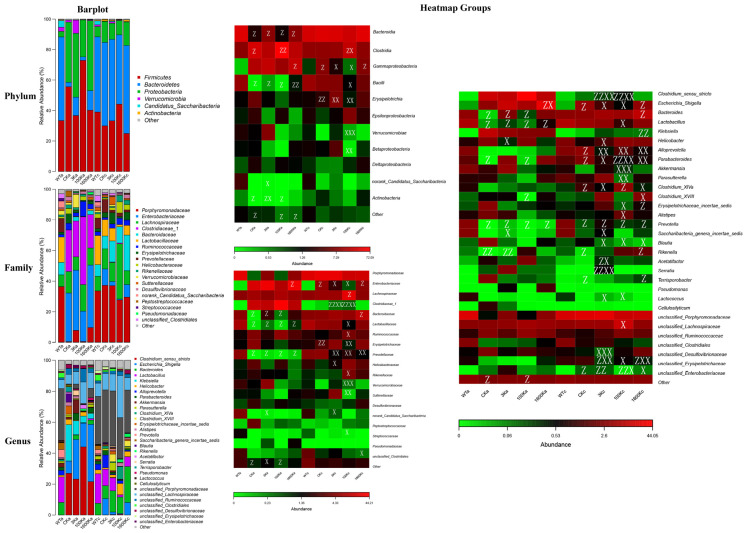
Effects of 3 kDA-HA,100 kDA-HA and 1600 kDA-HA on inter-group differences in mice intestinal microbiota and the abundance of dominant bacteria at different taxonomic levels were investigated at day 30. Community structure at different taxonomic levels and heatmap results of dominant species at different taxonomic levels are shown. ^z^
*p* < 0.05, ^zz^ *p* < 0.01 versus WT group; ^x^
*p* < 0.05, ^xx^ *p* < 0.01 or ^xxx^
*p* < 0.001 versus CK group.

## Data Availability

If the data supporting reported results are required, please contact the authors.

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
