# Peer review of "High-Molecular-Weight Hyaluronic Acid Can Be Used as a Food Additive to Improve the Symptoms of Persistent Inflammation, Immunosuppression and Catabolism Syndrome (PICS)"

_biology, 2024, doi:10.3390/biology13050319_

Round 1
Reviewer 1 Report
Comments and Suggestions for Authors
The manuscript titled: High molecular weight HA can be used as a food additive to improve the symptoms of PICS. This work studies the use of high molecular weight HA to improve the symptoms of PICS. The manuscript is well-structured and written. The methodology is pertinent, the results are interesting and demonstrate the HA potential effect alleviating PICS.
I recommend it for publication after minor changes.
Some observations about the manuscript are:
1. I suggest to improve the introduction (starting with PICS description and describing better the HA effect in the intestinal system and microbiota).
2. The results section is hard to follow in some sections, please consider to write a brief methodology description in each paragraph or the objective of each one).
3. In the results section: Explain the meaning of WT, CK, 3K, 100K, and 1600K
4. Legends in Figures 4B and 4C are too small; the same in Figure 5
5. Although in this study the beneficial effect of High molecular weight HA against the symptoms of PICS is proved; in the manuscript, the argumentation about the potential use of this molecule as a food additive is weak.
6. In the discussion section: I suggest to add more information about HA as a food’s additive (It is common to use HMW HA as an additive in food?).
Reviewer 2 Report
Comments and Suggestions for Authors
Based on a detailed review of your academic paper "High molecular weight HA can be used as a food additive to improve the symptoms of PICS," here are specific areas that require adjustments:
Title: I suggest to incorporate the animal model into the title for clarity, e.g., "Effect of High Molecular Weight Hyaluronic Acid on PICS Symptoms in a Mouse Model."
Abstract: The abstract should mention the key outcome that how high molecular weight HA significantly improved PICS symptoms compared to other molecular weights. Some concrete experimental data should be given as support.
Introduction
A brief overview of previous findings on HA's molecular weights affecting biological activities should be included, especially related to PICS, to provide a stronger rationale for focusing on high molecular weight HA.
Discussion
Please strengthen the discussion by directly linking your findings to potential clinical applications and addressing the translational aspect of using high molecular weight HA in human dietary interventions.
Conclusion
Please emphasize the superiority of high molecular weight HA in managing PICS symptoms effectively. The limitations of this study should be pointed out.
Comments on the Quality of English Language
Moderate editing of English language required
Author Response
Dear reviewer:
Thank you for your comments. We have studied comments carefully and have made correction which we hope meet with approval. Revised portion are marked in red in the revision, and the point-by-point response to the comments are as following:
Title: I suggest to incorporate the animal model into the title for clarity, e.g., "Effect of High Molecular Weight Hyaluronic Acid on PICS Symptoms in a Mouse Model."
Response 1: Thank you for your suggestion. However, since the original purpose of this study was to investigate the therapeutic effect of different molecular weight especially medium molecular weight HA in PICS mouse models, and the final result is that high molecular weight HA is better. Therefore, we hope that you can accept our original topic.
Abstract: The abstract should mention the key outcome that how high molecular weight HA significantly improved PICS symptoms compared to other molecular weights. Some concrete experimental data should be given as support.
Response 2: Thank you for your suggestion. The abstract has been rewritten. line 20-24.
Introduction
A brief overview of previous findings on HA's molecular weights affecting biological activities should be included, especially related to PICS, to provide a stronger rationale for focusing on high molecular weight HA.
Response 3: Thank you for your suggestion. The introduction has been rewritten. line 42-57
Discussion
Please strengthen the discussion by directly linking your findings to potential clinical applications and addressing the translational aspect of using high molecular weight HA in human dietary interventions.
Response4: Thank you for your suggestion. The discussion has been rewritten. line 264-272.
Conclusion
Please emphasize the superiority of high molecular weight HA in managing PICS symptoms effectively. The limitations of this study should be pointed out.
Response: Thank you for your suggestion. The limitations have been added. line 259-272.
Reviewer 3 Report
Comments and Suggestions for Authors
The article "High molecular weight HA can be used as a food additive to improve the symptoms of PICS" is very interesting and well written. In this article they have investigated different molecular weights of hyaluronic acid (HA) as a functional food additive against the amelioration of persistent inflammation, immunosuppression and catabolism syndrome (PICS). And they have found that high molecular weight (HMW) HA significantly improved PICS in mice and greatest potential to be an adjunct in alleviating PICS as a food additive. The introduction of the article was well written but can be improved by reviewing the recent literature on PICS.
Figure 1: please explain WT, CK, etc. in the legend.
Figure 1C and D: looks confusing, instead of comparing different p-values, please do a Tukey HSD test to show that they are significantly different.
Line 49-50: "Typical HMW HA (1600kDa), MMW HA (100kDa), and LMW HA (3kDa) were administered to this mouse model". In this case, I would suggest giving the range of molecular weight and giving the number average and weight average molecular weight etc. Have you tested them?
Figure 4 and 5: The figure legends are not clearly visible, you may not have enough space to show them. In this case, include them in the supporting information.
Comments on the Quality of English Language
English language of the article is fine and only minor editing is required.
Reviewer 4 Report
Comments and Suggestions for Authors
The authors of the manuscript presented research on persistent inflammation, immunosuppression and catabolism syndrome (PICS) in mouse models by hyaluronic acid (HA). The results of pathological observations and gut flora showed that medium-molecular-weight HA rapidly eliminated lung lesions and intestinal structural changes in PICS mice in the short term with reduced proportions of harmful bacteria in gut flora in the long term.
There is a gap in the literature regarding the research presented in the manuscript. The methodology is well-defined. The conclusions are consistent with the evidence and arguments presented. Any additional comments on the tables and figures and the quality of the data and text in the manuscript are presented below:
1. Please include the full names of HA and PICS in the article title.
2. Correct text in line 7.
3. Line 34 - improve the number of references [33,34] to [10,11]. Check the entire text and correct the reference numbers.
4. Line 36- write the capital letter D in the chemical compound named “β-1,4-d-glucuronic acid”.
5. Line 38 – add the space before references for example: „safety[32]”. Check the entire text and correct it.
6. Line 40 and line 211 - Latin names are written in italics, e.g. “in vivo”.
7. Line 42 – delete the space between “ than LMW”, line 45 – delete the space between “Thus,it “, etc. Check the entire all text and correct it.
8. Line 49 – “Amanda et al. [14].” It is not a correct reference. Check the entire text and correct the references. “1600kDa”is the number and the unit should be written separately. Check the entire text and correct them.
9. Figure 1 – Correct the “3kDA” and subsequent molecular weights. What means the “100 K, 1600 K, 3K, CK, WT” – Abbreviations need to be explained.
10. Page 4 – before line 126 add the text from page 5.
11. Figure 4 should be bigger, the legends of Figures 4A, 4B and 4C are not visible.
12. Latin names of plants are written in italics (lines 134, 135, 137,186-190, etc.). Please, check and correct them.
13. Figure 5 should be bigger, the legends of Figure 5 are not visible.
14. Line 152 - - correct the text to normal, not bold.
15. Line 159 – correct the word “Oue” and continue the sentence before the word “that”.
16. Line 164- delete the redundant dot.
17. Line 166- add the space between “both.Thus”
18. Line 168 – consider changing the word “Has”.
19. Line 175 – correct the words/sentence “of Has”.
References.
a) Write the references including the write the author’s name and abbreviation of the journal name (lines: 295 and next references) according to the instruction for the author’s Biology. Please, check and correct them.
b) Latin names of plants are written in italics (lines 339,364, etc.). Please, check and correct them.
c) The title of the article is written in lowercase letters, apart from the first word and proper names (lines 295,298, etc.). Please, check and correct them.
Conclusion: I recommend this manuscript be published in Biology after minor revision.
Comments on the Quality of English Language
Minor editing of English language required.
Author Response
Dear reviewer:
Thank you for your precious comments and advice. Those comments are all valuable and very helpful for revising and improving our paper, as well as the important guiding significance to our research. We have studied comments carefully and have made correction which we hope meet with approval. Revised portion are marked in red in the revision. The responds to the comments are as flowing:
The authors of the manuscript presented research on persistent inflammation, immunosuppression and catabolism syndrome (PICS) in mouse models by hyaluronic acid (HA). The results of pathological observations and gut flora showed that medium-molecular-weight HA rapidly eliminated lung lesions and intestinal structural changes in PICS mice in the short term with reduced proportions of harmful bacteria in gut flora in the long term.
There is a gap in the literature regarding the research presented in the manuscript. The methodology is well-defined. The conclusions are consistent with the evidence and arguments presented. Any additional comments on the tables and figures and the quality of the data and text in the manuscript are presented below:
- Please include the full names of HA and PICS in the article title.
Response 1: Thank you for your suggestion. The information has been added.
- Correct text in line 7.
Response 2: Sorry for this error,we have corrected it.
- Line 34 - improve the number of references [33,34] to [10,11]. Check the entire text and correct the reference numbers.
Response 3: Sorry for this error,we have re-proofread the number of references for the entire text.
- Line 36- write the capital letter D in the chemical compound named “β-1,4-d-glucuronic acid”.
Response 4: Sorry for this error, we have corrected it.
- Line 38 – add the space before references for example: „safety[32]”. Check the entire text and correct it.
Response 5: We have corrected these errors.
- Line 40 and line 211 - Latin names are written in italics, e.g. “in vivo”.
Response 6: We have corrected it.
- Line 42 – delete the space between “ than LMW”, line 45 – delete the space between “Thus,it “, etc. Check the entire all text and correct it.
Response 7: We have corrected these two errors.
- Line 49 – “Amanda et al. [14].” It is not a correct reference. Check the entire text and correct the references. “1600kDa”is the number and the unit should be written separately. Check the entire text and correct them.
Response 8: This reference has been deleted. We have also checked the entire text to correct this error.
- Figure 1 – Correct the “3kDA” and subsequent molecular weights. What means the “100 K, 1600 K, 3K, CK, WT” – Abbreviations need to be explained.
Response 9: These errors have been corrected. The information have been added. We have added the meaning in the Figure 1 legend and the body of manuscript (line 287-299).
- Page 4 – before line 126 add the text from page 5.
Response 10: Thank you for your suggestion. However, because we have added some contents and modified some Figures according reviewers’ comments, the typesetting would be poorly constructed if placed these text on one page. We hope that you will make a decision after reading the revision. If you think it still needs to be revised, we will revise it again.
- Figure 4 should be bigger, the legends of Figures 4A, 4B and 4C are not visible.
Response 11 :This Figure was revised.
- Latin names of plants are written in italics (lines 134, 135, 137,186-190, etc.). Please, check and correct them.
Response 12 :These mistakes has been corrected.
- Figure 5 should be bigger, the legends of Figure 5 are not visible.
Response 13:This Figure was re-graphed.
- Line 152 - - correct the text to normal, not bold.
Response 14 :We have corrected this error according your comment
- Line 159 – correct the word “Oue” and continue the sentence before the word “that”.
Response 15 :This mistake has been corrected.
- Line 164- delete the redundant dot.
Response 16 :This mistake has been corrected.
- Line 166- add the space between “both.Thus”
Response 17 :This mistake has been corrected.
- Line 168 – consider changing the word “Has”.
Response 18 :This mistake has been corrected.
- Line 175 – correct the words/sentence “of Has”.
Response 19 :This mistake has been corrected.
References.
a)Write the references including the write the author’s name and abbreviation of the journal name (lines: 295 and next references) according to the instruction for the author’s Please, check and correct them.
Response :This mistake has been corrected.
b)Latin names of plants are written in italics (lines 339,364, etc.). Please, check and correct them.
Response :This mistake has been corrected.
c)The title of the article is written in lowercase letters, apart from the first word and proper names (lines 295,298, etc.). Please, check and correct them.
Response :This mistake has been corrected.